# Study on Welding Deformation and Optimization of Fixture Scheme for Thin-Walled Flame Cylinder

**DOI:** 10.3390/ma15186418

**Published:** 2022-09-15

**Authors:** Yi Li, Yihao Li, Xiuping Ma, Xuhao Zhang, Dingyao Fu, Qitong Yan

**Affiliations:** 1School of Mechanical Engineering, Xiangtan University, Xiangtan 411105, China; 2Aecc South Industry Company Limited, Zhuzhou 412000, China

**Keywords:** welding deformation, TIG, clamping conditions, PSO

## Abstract

In this paper, the best fixture scheme for the TIG welding torch of nickel-base solid solution superalloy GH3536 in the welding process is explored. First of all, to meet the extremely high-dimensional accuracy requirements of the flame cylinder, a multifield coupling analysis model based on the flame cylinder is established on SYSWELD software. By studying the stress and deformation of welded parts under different line constraint positions and applied pressure, the trend of welding deformation is obtained, and the relevant mathematical model is established based on this. Finally, the particle swarm optimization (PSO) algorithm is used to calculate the best fixture scheme to make the welding stress and deformation better. The simulation results show that the welding deformation is negatively related to the line constraint distance and positively related to the applied pressure. According to the optimized clamping scheme of PS0, through simulation calculation, the average axial deformation is reduced by 82.5%, the maximum radial shrinkage deformation is reduced by 60.6%, and the maximum residual stress is reduced by 60.3%. Finally, it is verified by the flame barrel experiment that it meets the acceptance requirements and successfully solves the problem of serious axial shrinkage during the TIG welding of the outer ring of the flame barrel.

## 1. Introduction

The combustion chamber is one of the most important components in the entire engine, and its design directly affects the working condition and service life of the entire engine. Therefore, the combustion chamber is required to have important characteristics, such as high energy utilization rate, stable energy supply, good temperature distribution performance, low pressure loss, reliable structure, light weight, and long service life, so as to ensure the safe and efficient normal operation of aerospace engines [1,2]. The welding process is a very complex thermal process. The improvement of the welding process through many welding experiments and comparative analysis not only has higher cost and longer period but also is disturbed to bring experimental errors by external factors, and it is difficult to grasp the changing laws of its thermal and mechanical processes [3,4,5,6]. Nowadays, with the advancement of computer technology, welding simulation technology has also developed to maturity [7,8]. The method of welding numerical simulation can not only effectively reduce the material and time cost of the test but also simulate the time change process of welding temperature, residual stress, and the post-weld deformation of the structure under different conditions in a short time. At the same time, the main factors affecting welding residual stress and welding deformation can be analyzed and studied with a high degree of freedom according to the actual situation. In addition, experimental errors caused by various off-site factors in the actual experiment process can also be avoided. Because of the welding heat source directly applied to the welding position, it will bring a large amount of concentrated heat output, and the temperature of the molten pool will be much higher than that of the adjacent position. The difference in volume change caused by the temperature gradient difference will cause residual stress and welding deformation after welding cooling, which affects the subsequent processing and use of the workpiece. The application of the welding fixture exerts a supporting and tightening effect on the workpiece, which can inhibit and offset part of the welding deformation [9]. Guo simulated actual clamping conditions by using displacement constraints to study the pulsed laser welding of a Hastelloy C-276 sheet and concluded that the welding residual deformation can be controlled by reducing the line constraint distance [10]. Ma used the local inherent deformation method to impose inherent deformation on the structure and studied the influence on welding deformation with or without constraints and found the influence law of restraint state on welding deformation [11]. Tang constructed a simulation model to study large-scale thin-walled structures through the stress mapping method and used the PSO to optimize the clamping conditions, and the overall strain energy and the welding deformation of the structure were significantly reduced after optimization [12]. Wang Chenxi established a multilayer and multichannel coupled analysis model and found that the initial restraint force affects the welding deformation [13]. It can be concluded from the above references that welding deformation can be effectively reduced by using displacement constraints and applying pressure. However, applying different displacement constraints and pressures to multiple groups for welding will consume a lot of resources. Through welding finite element simulation technology, a large number of experiments required for the development of flame cylinders can be replaced to a certain extent, saving time, manpower, and material resources. Therefore, this paper uses SYSWELD software to simulate the TIG welding process of flame cylinders and studies the influence of different positional line constraints and different applied pressures on the deformation trend of the flame cylinder according to the defects of the original fixture. Finally, the Kriging method is used to obtain the fitness evaluation function, and the particle swarm optimization algorithm is used to calculate the best clamping scheme to make the welding stress and deformation better. The final calculation results are compared with the experimental results to verify the validity of the calculation results.

## 2. Numerical Simulation Calculation

### 2.1. The Finite Element Model of the Flame Cylinder

In this experiment, one of the weld seams of the flame cylinder connected by TIG was taken as the object, the wall thickness of the workpiece was 0.6 mm, and the diameter of the weld was about 180 mm. The wall thickness was thin, and there was no groove design during welding. Based on the shape characteristics of the outer rings, the use of three-dimensional solid mesh not only had too many meshes but also made it difficult to solve the problem of excessive mesh aspect ratio. Therefore, a two-dimensional shell finite element model was established and meshed to take calculation accuracy and efficiency into account. The model used a visual environment to mesh the model and adopted nonuniform mesh density transition. There were a total of 9691 2D meshes in this model, and the elements with Jacobian below 0.6 only accounted for 0.01%, which means that the mesh quality is good. The overall shape of the mesh is shown in Figure 1.

### 2.2. Establishment of Material Database for Nickel-Base Superalloy GH3536

As a typical solid solution superalloy, GH3536 is used widely in aviation production [14]. Its main solid solution-strengthening elements are Mo, W, and Co, and it has good corrosion resistance, oxidation resistance, and strength at high temperature [15,16]. In addition, its hot and cold processing performance and welding performance also perform well. Referring to the China Aviation Materials Manual, its main elements are in Table 1 as follows.

The thermal conductivity, specific heat, and density distribution curve of GH3536 were obtained by consulting reference [17]. The yield strength and stress–strain curves at various temperatures were obtained through high-temperature tensile tests and processed to obtain work hardening curves. Figure 2 shows the main performance parameters of GH3536.

### 2.3. Selection of Heat Source Model

The Goldak double ellipsoid heat source model is used in this paper. Due to the different lengths of the front and rear axles, the model can accurately reflect the different characteristics of the temperature gradient before and after the molten pool during the welding process. At the same time, as a body heat source, it can also describe the continuous decay of heat along the depth direction of the molten pool [18,19]. It is widely used in the temperature field simulation of various welding methods such as TIG, MIG, MAG, etc. The heat flux density distribution inside the ellipsoid of the Goldak double ellipsoid heat source model is divided into two parts: the front part and the back part, respectively [20]:(1)$$q_{b}\left(x,y,z\right)=\frac{6\sqrt{3}f_{b}\left(\eta UI\right)}{\pi \sqrt{\pi }a_{b}bc}\exp\left[-3\left(\frac{x^{2}}{a_{b}^{2}}+\frac{y^{2}}{b^{2}}+\frac{z^{2}}{c^{2}}\right)\right],x>0$$
(2)$$q_{f}\left(x,y,z\right)=\frac{6\sqrt{3}f_{f}\left(\eta UI\right)}{\pi \sqrt{\pi }a_{f}bc}\exp\left[-3\left(\frac{x^{2}}{a_{f}^{2}}+\frac{y^{2}}{b^{2}}+\frac{z^{2}}{c^{2}}\right)\right],x>0$$

In the formula, $f_{b}$ and $f_{f}$ are the heat ratios of the front and rear halves, respectively; $f_{b}+f_{f}=1$; η is the arc thermal efficiency; *U* is arc voltage; *I* is welding current; and $a_{b},a_{f},b,$ and *c* are the shape parameters of the double ellipsoid.

## 3. Experiment

The experiment used nonmelting electrode gas-shielded welding (TIG) with argon as the shielding gas to connect the thin-walled flame cylinder. The flame cylinder was made of nickel-base superalloy GH3536. Figure 3 shows the structure of the flame cylinder. The wall thickness was 0.6 mm, argon was used as the shielding gas, and the welding wire was not filled. The schematic diagram of the welding fixture is shown in Figure 4.

Before welding, the surface of the weld area was cleaned, and the acetone-solution on the surface was wiped to avoid the influence of oil and impurities on the surface on the welding results; the butt gap was minimized as much as possible and the weld was fixed by spot welding, which can effectively avoid defects such as dislocation. A total of 2 samples was produced, and the specific welding parameters are as follows in Table 2. The height was measured after the weldment cooled to room temperature. From the initial point of the weld seam, 8 measuring points were evenly distributed to measure the size of the workpiece to obtain the overall axial deformation of the flame cylinder after TIG.

After the TIG welding was completely cooled, we measured the height. From the initial point of the weld, 8 measuring points were evenly distributed to measure the workpiece size and obtain the overall axial deformation of the TIG welding. In the welding process, the axial and radial deformation of the sample are not strictly controlled parameters, but the height of the sample after welding is the core parameter. Therefore, only the welding height was measured, as shown in Figure 5. In order to verify the accuracy of the simulation results, two workpieces were welded, named as part 1 and part 2, respectively. The simulation results, called part 0, were compared with the welded parts 1 and 2 in the height direction, as shown in Table 3.

Figure 6 shows the comparison between the actual value and the simulated value of the overall axial deformation of the two flame tube parts. The simulation and actual measurement results are both shrinking to the weld seam. The simulation deformation results are generally consistent with the deformation of part 1, indicating that the simulation effect is good. The simulation average axial shrinkage of 0.52 mm is like the experimental average axial shrinkage of 0.46 mm and 0.48 mm, and the error is 9.6%.

## 4. Influence of Clamping Conditions on Welding Stress and Deformation

### 4.1. Influence of Line Restraint Distance on Welding Stress and Deformation

In this section, we explain that the weldment was positioned and clamped by a ring clamp parallel to the weld seam, and the fixed position could be moved in the direction perpendicular to the weld seam to select the optimal clamping position through welding simulation calculation. Combined with the actual production situation of the flame cylinder, the minimum application distance of the line constraint should not be less than 5 mm, and the maximum distance should not exceed 40 mm.

The results of welding stress and deformation caused by different line constraint distances were selected, and the deformation curve law was obtained, as shown in Figure 7. The axial deformation trends obtained by applying different positional line constraints are basically similar, and the farther the positional constraint is from the weld position, the larger the welding axial shrinkage, radial shrinkage, and maximum residual stress. With the increase in the linear confinement distance, the average axial deformation of the flame barrel increases, but it decreases when the confinement distance exceeds 30 mm. When the linear constraint distance increases from 20 mm to 30 mm, the axial deformation of flame barrel welding is greatly affected. The average axial deformation increases rapidly, from 0.150 mm to 0.291 mm, with an increase of 94%, and the maximum residual stress decreases from 754 MPa to 502 MPa, with a decrease of 33%. Therefore, if the axial deformation of the outer ring of the flame barrel is to be controlled, the clamping mode with line constraint close to the weld should be selected as far as possible.

### 4.2. Influence of Applied Pressure on Welding Stress and Deformation

In this subsection, we explain the influence of the applied pressure on the welding stress and deformation of the flame cylinder components which was studied by applying applied pressures of different sizes, at the line restraint position, 10 mm perpendicular to the weld position, and 2 mm in width. The applied pressure was present throughout the welding process until the workpiece cooled to room temperature and was unloaded with the welding fixture at 800 s.

The simulation results of applying external pressure near the line constraint at a distance of 10 mm from the vertical weld are shown in Figure 8, and the application of appropriate applied pressure can reduce the axial deformation and greatly improve the residual stress distribution. However, the radial deformation will increase, and when the applied pressure exceeds 60 MPa, the structure will cause buckling distortion under the action of excessive pressure. The stress concentration at the applied position of the applied pressure is obvious, and large plastic deformation will occur, resulting in a rapid increase in the maximum residual stress.

## 5. Optimization of Clamping Conditions

### 5.1. Determination of Comprehensive Evaluation Function

According to the above, the line restraint and applied pressure at different positions affect the welding stress and deformation of the workpiece to different degrees [21]. In this paper, the optimal welding clamping combination is obtained by selecting different position line constraints and applied pressures, and a corresponding multiobjective optimization model is established for the model analysis of the flame cylinder components. The influence factors were obtained by assigning different weights to welding residual stress and deformation and summing them linearly to obtain a comprehensive fitness evaluation function. In the optimization objective, different weight coefficients were assigned according to the order of magnitude of the objective and it was determined that the weight coefficient of welding residual stress $\sigma_{max}$ was 1, the weight coefficient of longitudinal deformation $U_{i1}$ was 1000, and the weight coefficient of lateral deformation $U_{i2}$ was 60. Finally, the optimized comprehensive evaluation function was:(3)$$g=1000U_{i1}+60U_{i2}+\sigma_{\max}$$

### 5.2. Determination of Kriging Model

The Kriging method [22,23], also known as the spatial local interpolation method, is an improvement in the linear regression of interpolation method. It uses the known raw data in the sample and does not depend on a specific mathematical model, and the optimal estimation of unknown nodes is achieved by combining its variogram. The Kriging model can fully consider the variation trend and characteristic attribute values of variables in its region by considering the information of known sample nodes and the situation of unknown nodes nearby with high local accuracy. Therefore, the Kriging method is much higher than the surface response method in both local and overall simulation accuracy, and the Kriging method does not require a specific mathematical model, making it easier to apply.

The Kriging prediction model was established by designing the comprehensive evaluation function of 24 groups of samples obtained by designing different line restraint application positions corresponding to the magnitude of the applied stress. The data are shown in Table 4, and the fitting plane of the Kriging model is shown in Figure 9. Finally, we obtained the regression results: θ=[20.0000,2.9730] and
β=[−0.1901;0.7265;0.4726;−0.1363;0.0725;0.3389].

### 5.3. Particle Swarm Optimization Algorithm (PSO)

#### 5.3.1. Theory of PSO

The PSO, the particle swarm optimization algorithm, is a random optimization technology based on population. It was proposed by Eberhart and Kennedy in 1995. The PSO selects the optimal solution from the results of multiple iterations through random solutions [24,25,26,27]. In PSO, the algorithm regards the possible solution in the solution range as a bird in the flock by simulating the predation behavior of the flock. Each bird knows the distance between its own position and the food, and all particles in the population are pointed to the position of a better possible solution through the information transfer between birds to gradually increase the probability of finding a better solution along the way.

Similar multiobjective optimization algorithms include the genetic algorithm (GA), simulated annealing algorithm (SAA), decision tree algorithm, and so on. In the process of optimization algorithms seeking the optimal solution, the PSO is simple to operate, is easy to implement, and has stronger global optimization capability and higher work efficiency, which have been widely used in dealing with nonlinear optimization problems. The overall scale of the PSO was set to N, and there was an M-dimensional search space for positioning. Each particle was initialized and the optimal position of the particle and the global optimal position were changed. The particle updated its velocity and position information through the formula after obtaining the individual optimal position and the global optimal position. The relevant formula is as follows [28,29]:(4)$$v_{j}^{i}\left(k+1\right)=w\left(k\right)v_{j}^{k}\left(k\right)+\varphi ,\mathrm{ran}d\left(0,a_{1}\right)\left(p_{j}^{i}\left(k\right)-x_{j}^{i}\left(k\right)\right)+\varphi_{2}\mathrm{rand}\left(0,a_{2}\right)\left(p_{j}^{g}\left(k\right)-x_{j}^{i}\left(k\right)\right)$$
(5)$$x_{j}^{i}\left(k+1\right)=x_{j}^{i}\left(k\right)+v_{j}^{i}\left(k+1\right)$$

#### 5.3.2. The Optimization Results of the Fixture Scheme of the Flame Cylinder

The particle swarm optimization algorithm was used to determine the optimal value of the applied position of the line constraint and the applied stress, so as to minimize the value of the obtained comprehensive evaluation function equation. Among them, the main setting parameters of the particle swarm were the number of particles in the particle swarm, the initial weight coefficient, the maximum iteration steps, and the learning factors 1 and 2. The parameter settings are shown in the following table.

The number of particles is generally 20–40, and the small population size easily falls into local optimization; a large population size can improve the convergence, but for most problems, 10 particles are enough to achieve good results. Therefore, we set the number of particles in the particle swarm to 10, and the number of iterations was usually 50–100. If the number is too small, instability will occur. Therefore, we set the maximum iteration step to 100. A larger weight coefficient is conducive to global search, while a smaller weight coefficient is conducive to local search, so that the algorithm can quickly converge to the optimal solution. Generally, the value range of the weight coefficient is 0.4–2. After comprehensive consideration, the initial weight coefficient was set to 0.8. The learning factor is also called the acceleration coefficient or acceleration factor. If the value is too low, the particles will wander outside the target area, and if the value is too high, the particles will cross the target area. The value range is usually 0–4. Since the target area of this model was small, the values of learning factor 1 and learning factor 2 were both 1. The optimization process was carried out with the objective of obtaining the optimal constraint application position and external stress. The optimization process is shown in the figure. With the increase in the number of iterations of the particle swarm optimization algorithm, the value of the objective function converges rapidly and fluctuates less in the subsequent iterative calculation and recalculates its adaptability with the population sizes of 20 and 30 to avoid falling into local optimization, as shown in Figure 10. The final optimization result is [5, 39.691], which corresponds to the numerical solution after the optimization of the clamp position and the applied pressure, respectively. The obtained optimal solutions are: 5 mm and 39.691 Mpa.

The welding simulation of the flame cylinder was rerun according to the optimized numerical solution of the clamp position and the applied pressure obtained by the PSO, bringing the obtained welding residual stress and deformation into the comprehensive evaluation function, and the obtained value was similar to the optimal solution obtained by the PSO.

The final result was selected as the line constraint distance of 5 mm from the weld and the applied pressure of 39.69 MPa. The optimized welding deformation and residual stress distribution results under this fixture scheme are shown in Figure 11. Comparing the fixture scheme before optimization, the welding stress and deformation of the flame cylinder after optimization have been significantly improved. The average axial deformation decreases by 82.5%, from 0.52 mm to 0.09 mm, the maximum radial shrinkage deformation decreases by 60.6%, from 1.27 mm to 0.50 mm; the maximum residual stress decreases by 60.3%, from 955.20 MPa to 378.89 MPa; and the optimization effect is good.

#### 5.3.3. Experimental Verification of the Optimized Fixture Scheme

The line constraint was applied at a vertical distance of 5 mm from the weld seam, and the applied pressure of 39.69 MPa was achieved by changing the initial clamping force of the fixture on the workpiece. The width of the bead was 2 mm, and the initial clamping force was about 46 kN. The axial average deformation of the flame cylinder components after welding was 0.17 mm, which is less than the 0.2 mm required by the process, as shown in Figure 12. The optimized fixture met the process requirements.

## 6. Conclusions

(1)The simulation results of different line constraint clamping schemes show that applying line constraints near the weld can effectively reduce the axial deformation of the outer ring of the flame cylinder and the highest residual stress. Applying different pressures near the line clamping constraint can reduce the axial deformation and improve the residual stress distribution, but the radial deformation will increase. When the applied pressure exceeds 60 MPa, the excessive applied pressure will cause instability and deformation in the thin-walled parts.(2)Through the particle swarm optimization algorithm, the clamping condition scheme with a comprehensive optimal evaluation of welding deformation and residual stress can be obtained: under the scheme of an online constraint distance of 5 mm and external pressure of 39.69 Mpa, the welding quality of the flame barrel is better, the average axial deformation is reduced from 0.52 mm to 0.091 mm, the axial shrinkage is reduced by 82.5%, and the maximum radial shrinkage is reduced by 60.6%, compared with the original clamping scheme. The maximum residual stress is reduced by 60.3%.(3)By changing the initial clamping force of the clamp on the workpiece, an external pressure of 39.69 MPa is achieved, in which the width of the bead is 2 mm and the distance from the weld is 5 mm. The initial clamping force is about 46 kn. The average axial deformation of the outer ring part of the flame barrel after welding is 0.17 mm, which is 63.0% less than that before the optimization of the clamping conditions and 0.2 mm less than the process requirements. The optimized clamp meets the process requirements.

## Figures and Tables

**Figure 1 materials-15-06418-f001:**
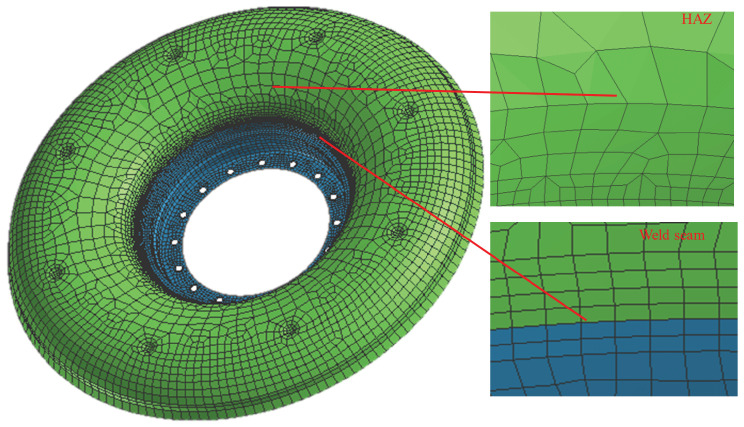
Diagram of meshing.

**Figure 2 materials-15-06418-f002:**
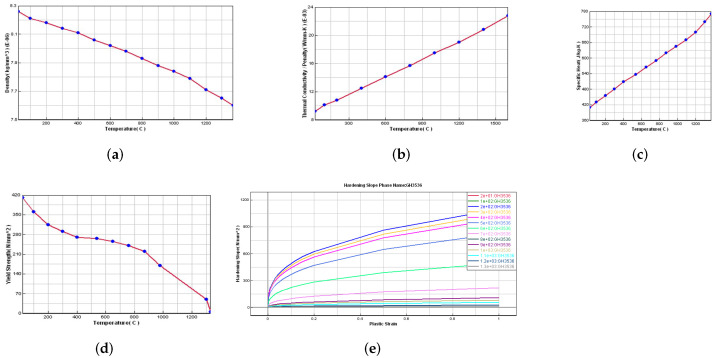
Diagram of material database for GH3536: (**a**) density, (**b**) thermal conductivity, (**c**) specific heat, (**d**) yield strength, and (**e**) stress–strain curves.

**Figure 3 materials-15-06418-f003:**
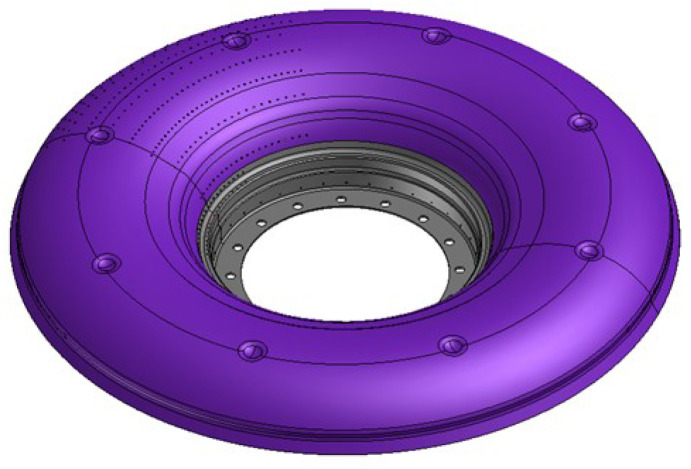
Schematic diagram of flame cylinder.

**Figure 4 materials-15-06418-f004:**
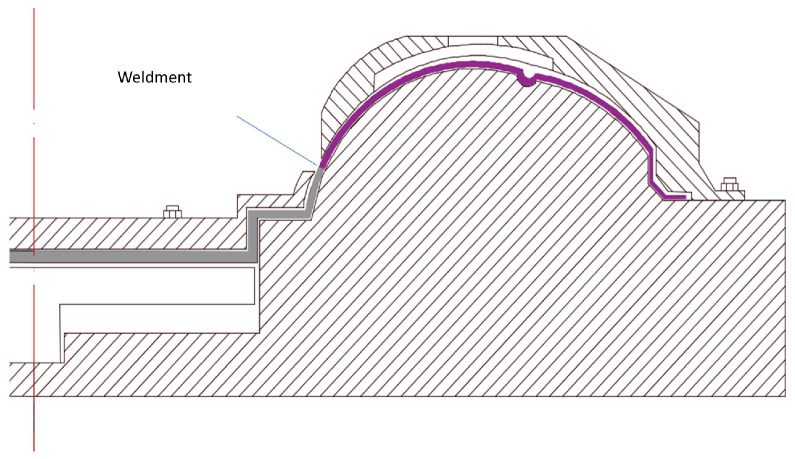
Schematic diagram of welding fixture.

**Figure 5 materials-15-06418-f005:**
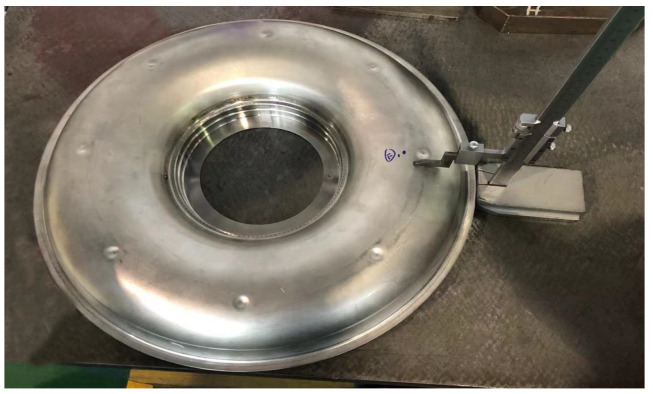
Schematic diagram of sample height measurement after welding.

**Figure 6 materials-15-06418-f006:**
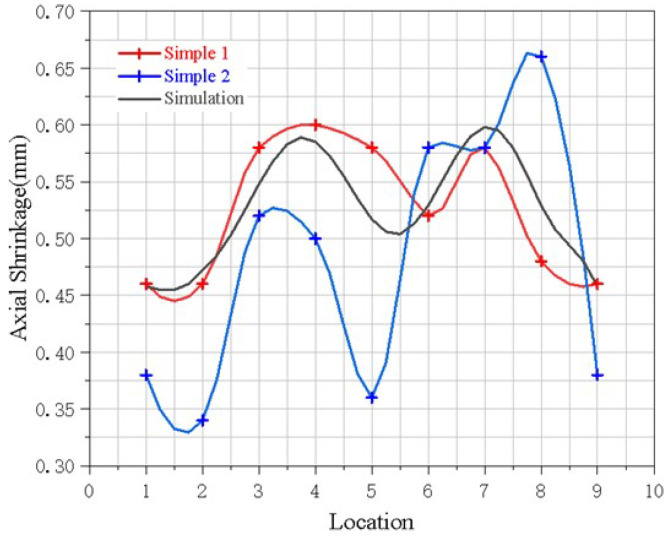
Welding Deformation.

**Figure 7 materials-15-06418-f007:**
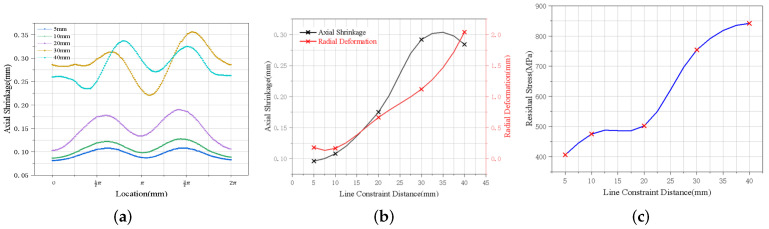
Stress and deformation at different line constraint distances: (**a**) deformation trend, (**b**) average axial shrinkage and radial deformation, and (**c**) residual stress.

**Figure 8 materials-15-06418-f008:**
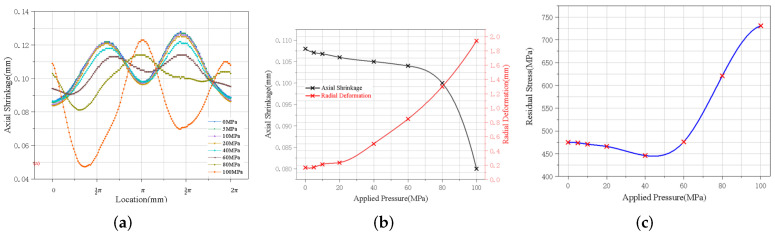
Stress and deformation at different applied pressures: (**a**) deformation trend, (**b**) average axial shrinkage and radial deformation, and (**c**) residual stress.

**Figure 9 materials-15-06418-f009:**
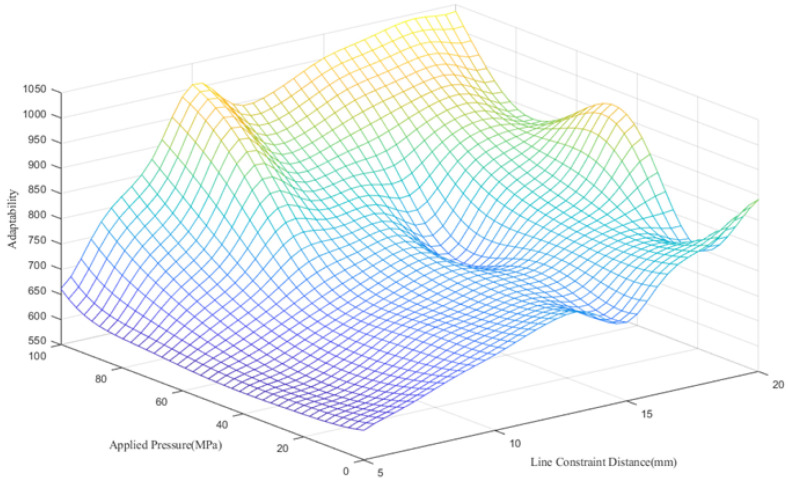
Diagram of the fitting plane of the Kriging model.

**Figure 10 materials-15-06418-f010:**
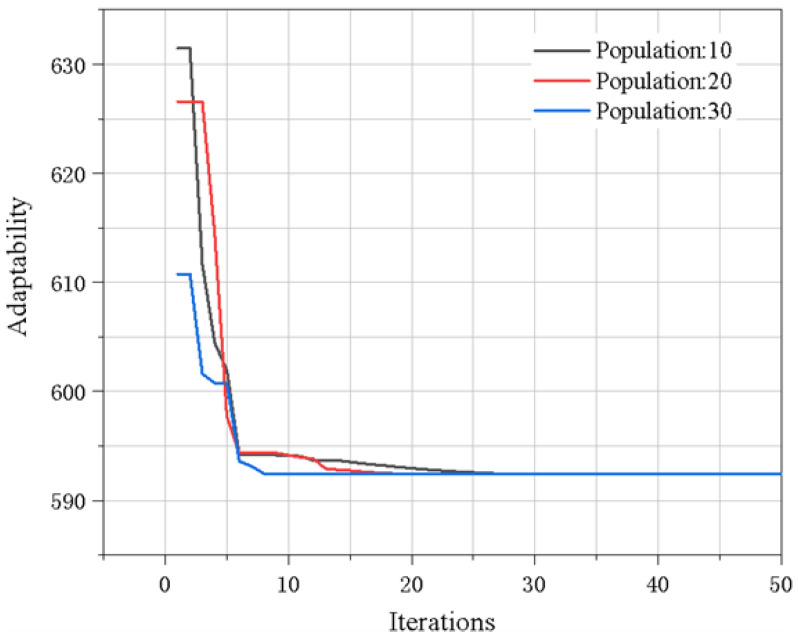
Iterative process of PSO.

**Figure 11 materials-15-06418-f011:**
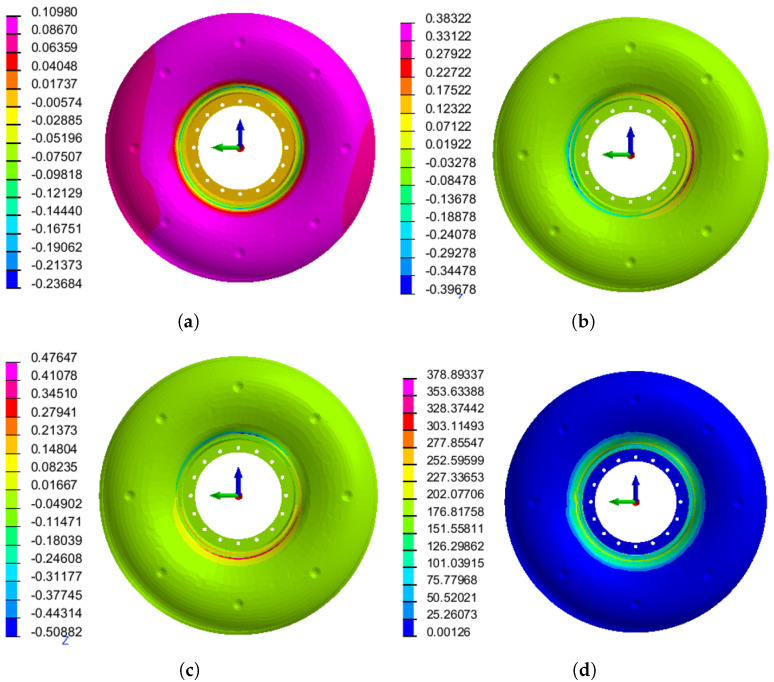
Simulation results: (**a**) axial deformation, (**b**) radial deformation (Y−axis), (**c**) radial deformation (Z−axis), and (**d**) residual stress.

**Figure 12 materials-15-06418-f012:**
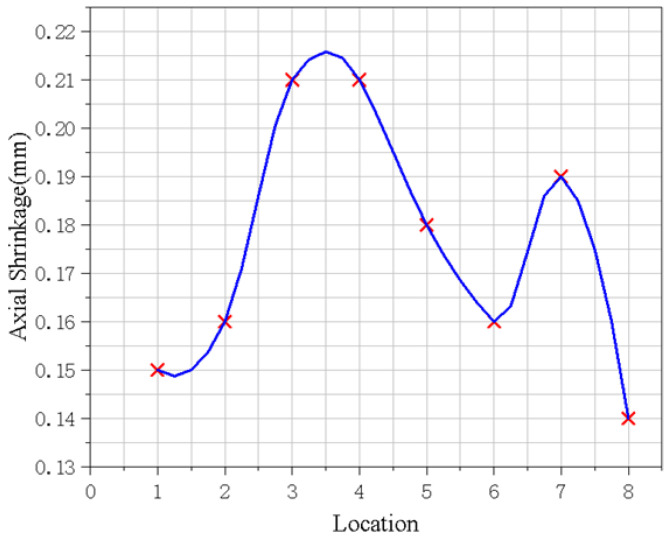
Axial shrinkage of experimental workpiece after optimization.

**Table 1 materials-15-06418-t001:** Chemical Composition of GH3536.

C	Cr	Ni	Co	W	Mo	Fe
0.05–0.15	20.5–23.0	remaining	0.50–2.50	0.20–1.00	8.0–10.0	17.0–20.0
**Al**	**Ti**	**B**	**Cu**	**Mn**	**Si**	**P**
			**Not more than**			
0.5	0.15	0.010	0.05	1.00	1.00	0.025

**Table 2 materials-15-06418-t002:** Specific welding parameters for TIG.

	Current (A)	Voltage (U)	Stick Out Distance (mm)	Welding Speed (mm/s)	Heat Input (KJ/mm)
**Sample 1**	17	10	1	2.14	0.079
**Sample 2**	16	12	2	2.14	0.090

**Table 3 materials-15-06418-t003:** Height comparison before and after welding.

		1	2	3	4	5	6	7	8	Average (mm)
**Part 0**	**Before**	66.3	66.3	66.3	66.3	66.3	66.3	66.3	66.3	66.3
	**After**	65.842	65.828	65.752	65.715	65.783	65.771	65.702	65.771	65.77
**Part 1**	**Before**	66.34	66.24	66.36	66.28	66.26	66.26	66.28	66.46	66.31
	**After**	65.88	65.78	65.78	65.68	65.68	65.74	65.7	65.98	65.78
**Part 2**	**Before**	66.32	66.2	66.1	65.98	66	66.1	66.18	66.22	66.13
	**After**	65.94	65.86	65.58	65.48	65.64	65.52	65.6	65.56	65.65

**Table 4 materials-15-06418-t004:** Comprehensive evaluation value corresponding to line constraint and applied pressure.

Distance (mm)	Pressure (MPa)	Axial (mm)	Radial (mm)	Residual Stress (MPa)	Value
5.00	0.00	0.10	0.18	406.41	609.24
5.00	20.00	0.09	0.25	393.45	595.43
5.00	40.00	0.09	0.50	378.89	592.40
5.00	60.00	0.09	0.88	363.57	598.07
5.00	80.00	0.09	1.37	346.69	608.97
20.00	80.00	0.11	2.54	507.26	889.38
20.00	100.00	0.11	3.15	618.59	1035.53

## Data Availability

Data available on request from the authors.

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
