# Peer review of "Study on Welding Deformation and Optimization of Fixture Scheme for Thin-Walled Flame Cylinder"

_materials, 2022, doi:10.3390/ma15186418_

Round 1

Reviewer 1 Report

Dear Authors,  

I had pleasure to review your paper titled "Study on welding deformation and optimization of fixture scheme for thin-walled flame cylinder".

The paper fulfils the aims and scope of Materials journal, and can be considereded for potential publication. Hoever, it needs some improvements. I have some suggestions, which are listed below. 

General remarks:

- Please add the quantitative results into the abstract.

Introduction:

- In this part, you should underline the necssity and novelty of your work. This aspect is not well presented. Whay new has been proposed? Why your work is important? Please underline these issues.

- As you stated, some factors are responsible for deformation of welded structures. You have described parameters, heat source, cooling etc.. However, e.g., tack weld disctubution is not mentioned. It has been tested for TIG welding.

Experimental:

- This part requires modifications. Firstly, there is no information about Base Material (BM). You should sho its chemcial compositon and basic mechanical propeties (yield point, tensile stregth, elongation). Each of them are responsible for properties of the joint, which are important from the deformation point of the view.

- Table 2 - what is "tungsten distance"? Please use welding engineering phrases. Moroever, the most important parameter is heat input [kJ/mm]. Please add this into the table.

You have presented only simulations. The calculations looks good. Hoever, never ever we will be sure, that they are good without real experiment. The comparision beteen simulation and real elded specimen allows to determine the conclusions. In my opinion, you should present the real experiment results too.

Reviewer 2 Report

In this work, the authors studied the welding deformation in an optimized fixture scheme for the thin-walled flame cylinder. The research appears to be efficiently done and appropriately reported, however, the standard of English is acceptable but only needs a few corrections. Nevertheless, some questions and corrections must be answered to improve and complete the manuscript.

Introduction section: In this section, the authors don’t indicate the novelty of their work. what is the innovation of your work when compared with the other researchers? The "Knowledge gap to be filled"? In this introduction, the authors must describe or indicate the work that will be done to test their "hypothesis".

Experiment section: This section is very poor and incomplete. The authors must describe in more detail the experimental work: Did the authors use a robot to weld the specimens? Which one? How did the authors the measurement of deformation? Did they use specific equipment? Which the error had the measurements? Did they measure the residual stresses?

Table 1. Please change “Tungsten distance (mm)” to “Stick-out (mm)” because it is a technical term.

Figure 4. Was this information obtained from experimental tests or from materials data baes? Which one?

Line 177. The authors wrote, “… in nature, people proposed PSO. The PSO… ”. Who proposed for the first time?

Lines 199-202. The authors wrote “… The number of particles in the particle swarm is set to 10, the maximum iteration step is 100, the initial weight coefficient is 0.8, and the learning factor 1 and learning factor 2 are set to 1…”. Why these values?

Line 203. The authors wrote, “… optimization process is shown in Fig.”. Which figure?

Conclusion: In the first paragraph the authors claimed that “Applying line restraint at the position near the weld seam can effectively reduce the axial shrinkage of the flame cylinder components and reduce the maximum residual stress value.”. Did you have another study that proves this statement or did you verify with experimental measurements?

Reviewer 3 Report

Study on welding deformation and optimization of fixture scheme for thin-walled flame cylinder

Article is interesting. Few observations are given below;

Some more latest studies are required in the introduction section to further highlight the importance of this study.

Suntharalingam, T., Upasiri, I., Nagaratnam, B., Poologanathan, K., Gatheeshgar, P., Tsavdaridis, K. D., & Nuwanthika, D. (2022). Finite Element Modelling to Predict the Fire Performance of Bio-Inspired 3D-Printed Concrete Wall Panels Exposed to Realistic Fire. Buildings, 12(2), 111.

Le, K. B., & Cao, V. V. (2021). Numerical Study of Circular Concrete Filled Steel Tubes Subjected to Pure Torsion. Buildings, 11(9), 397.

Section 2, is it experiment? or finite element analysis, if this is only analysis, then authors should not say experiment, please revise article carefully.

What was the reason to select this type of welding and metal.

Table 1 what is tungsten distance?

Figure 2, what authors indent to show? should revised with labels.

Section 3.1, how about mesh sensitivity analysis, how do authors selected the mesh size?

Figures 4-6, axis are not readable alongwith legends.

Authors must summarized results in more systematic way with reference to the previous studies.

Also, Conclusions are too limited to proof the significant outcome of this study.

Round 2

Reviewer 1 Report

Dear Authors,

Some issues still need improvements:

1. Table 1 - the source of presented values is unknown. Have you measured/tested these values? If yes, please describe methodology. If not, please show the source - standard, manufacturer data etc.,

2. Table 2 - following welding engineering standards, the unit for heat input should be [kJ/mm].

3. Fig. 5 has not been mentioned in the text
